# Accurate Screening for Early-Stage Breast Cancer by Detection and Profiling of Circulating Tumor Cells

**DOI:** 10.3390/cancers14143341

**Published:** 2022-07-09

**Authors:** Timothy Crook, Robert Leonard, Kefah Mokbel, Alastair Thompson, Michael Michell, Raymond Page, Ashok Vaid, Ravi Mehrotra, Anantbhushan Ranade, Sewanti Limaye, Darshana Patil, Dadasaheb Akolkar, Vineet Datta, Pradip Fulmali, Sachin Apurwa, Stefan Schuster, Ajay Srinivasan, Rajan Datar

**Affiliations:** 1Department of Oncology, The London Clinic, London W1G 6BW, UK; 2Department of Oncology, Cromwell Hospital, London SW5 0TU, UK; robert.leonard@nhs.net; 3The London Breast Institute, Princess Grace Hospital, London W1U 5NY, UK; kefah.mokbel@hcahealthcare.co.uk; 4Division of Surgical Oncology, Baylor College of Medicine, Houston, TX 77030, USA; alastair.thompson@bcm.edu; 5National Breast Screening Training Centre, King’s College Hospital, London SE5 9RS, UK; michael.michell@nhs.net; 6Department of Biomedical Engineering, Worcester Polytechnic Institute, Worcester, MA 01609, USA; rpage@wpi.edu; 7Department of Medical and Haemato Oncology, Medanta-The Medicity, Gurugram 122001, India; akvaid@yahoo.com; 8Rollins School of Public Health, Emory University, Atlanta, GA 30322, USA; ravi.kumar.mehrotra@emory.edu; 9Department of Medical Oncology, Avinash Cancer Clinic, Pune 411030, India; draaranade@gmail.com; 10Department of Medical and Precision Oncology, Sir HN Reliance Foundation Hospital and Research Centre, Mumbai 400004, India; sewanti.limaye@rfhospital.org; 11Department of Research and Innovations, Datar Cancer Genetics, Nasik 422010, India; drdarshanap@datarpgx.com (D.P.); dadasaheb.akolkar@datarpgx.com (D.A.); drvineetdatta@datarpgx.com (V.D.); pradipfulmali@datarpgx.org (P.F.); sachinapurwa@datarpgx.org (S.A.); ajays@datarpgx.org (A.S.); rajandatar@datarpgx.com (R.D.); 12Department of Research and Innovations, Datar Cancer Genetics Europe GmbH, 95488 Eckersdorf, Germany; drstefanschuster@datarpgx.com

**Keywords:** breast cancer, screening, circulating tumor cells, immunocytochemistry

## Abstract

**Simple Summary:**

Detection of breast cancer in the early stages is associated with higher cure rates and better survival, and also requires fewer intensive treatments. Current breast cancer screening via mammography is unsuitable for use among (younger) women with more dense breasts, and also has limitations in its ability to detect aggressive breast cancers. In this research article, we describe a breast cancer detection test that is based on the detection of ‘circulating tumor cells’ in blood samples. This test can detect breast cancer CTCs with high accuracy across all age groups, hormone receptor subtypes, histological subtypes, and disease grade. In our study, this test detected breast cancer cases and differentiated them from healthy (cancer-free) females as well as those with non-cancerous conditions with high accuracy. This test has negligible risk of false positive findings, as well as high detection rate for early-stage (localized) breast cancer. Clinical adoption of this test can be beneficial in cancer screening as well as in detection of breast cancers in suspected cases.

**Abstract:**

Background: The early detection of breast cancer (BrC) is associated with improved survival. We describe a blood-based breast cancer detection test based on functional enrichment of breast-adenocarcinoma-associated circulating tumor cells (BrAD-CTCs) and their identification via multiplexed fluorescence immunocytochemistry (ICC) profiling for GCDFP15, GATA3, EpCAM, PanCK, and CD45 status. Methods: The ability of the test to differentiate BrC cases (*N* = 548) from healthy women (*N* = 9632) was evaluated in a case–control clinical study. The ability of the test to differentiate BrC cases from those with benign breast conditions was evaluated in a prospective clinical study of women (*N* = 141) suspected of BrC. Results: The test accurately detects BrAD-CTCs in breast cancers, irrespective of age, ethnicity, disease stage, grade, or hormone receptor status. Analytical validation established the high accuracy and reliability of the test under intended use conditions. The test detects and differentiates BrC cases from healthy women with 100% specificity and 92.07% overall sensitivity in a case–control study. In a prospective clinical study, the test shows 93.1% specificity and 94.64% overall sensitivity in differentiating breast cancer cases (*N* = 112) from benign breast conditions (*N* = 29). Conclusion: The findings reported in this manuscript support the clinical potential of this test for blood-based BrC detection.

## 1. Background

Breast cancer (BrC) is the most common malignancy, and a leading cause of cancer-related mortality among women globally [1]. Although mammography is the standard of BrC screening in asymptomatic females, there is a need for improved BrC detection which addresses the risks and limitations of mammography, such as radiation exposure, lower specificity in differentiating benign conditions from malignancy, and lower sensitivity for invasive carcinomas, as well as incompatibility with dense breast tissue. Circulating tumor analytes in peripheral blood were evaluated for potential application in more accurate, non-radiological, and non-/minimally invasive screening for breast cancer. Circulating tumor cells (CTCs) are an ideal analyte for detection of cancers, since they are intact malignant cells that harbor the imprint of the parent tumor. CTCs have distinct advantages over nucleic acid fragments or serum antigens, since the latter may also be released by non-malignant cells, and are associated with lower sensitivity and specificity, respectively. There is evidence of sufficient viable CTCs being released into blood even during the early stages of carcinogenesis. In breast cancer, it is reported that angiogenesis commences at the DCIS stage itself, which can facilitate the dissemination of tumor cells [2,3,4]. Such disseminated tumor cells (DTCs) are reported in bone marrow of 20% to >50% of patients with DCIS or DCIS with microinvasion, respectively [5,6,7]. Prior studies also indicate high detection rates of CTCs in blood samples of patients with early-stage breast cancers. Using nanostructured coated slides, Krol et al. [8] report 62.5% CTC detection rate for stage I and II BrC. Using filtration-based devices, Reduzzi et al. [9] show a 76% CTC detection rate in early-stage breast cancer. Similarly, Jin et al. [10] use the CytoSorter^®^ CTC capture system, and show 50% and >80% sensitivity in DCIS and stage I/II BrC. Fina et al. report >78% CTC detection rates in early-stage breast cancers using an antigen-independent method [11], and 65% CTC detection rate using antigen-dependent (EpCAM, ERBB2, and EGFR expression) capture, followed by quantitative polymerase chain reaction (qPCR) profiling of targeted gene panel [12]. These studies support the biological plausibility of CTC-based cancer screening approaches. Although CTCs were evaluated for cancer detection, the inability of prior technologies to effectively enrich and harvest sufficient CTCs hindered meaningful downstream applications. Most prior attempts at evaluating CTCs for cancer screening were based on epitope capture using the CellSearch platform, which, while not approved for CTC detection, is frequently used in research. Several prior studies highlight the lower performance of epitope capture, arising due to its inability to efficiently harvest or detect CTCs with lower expression of EpCAM and PanCK, which are the most routinely employed target markers [13,14,15,16,17,18,19], with some improvements in sensitivity when epitope capture is used in combination with gene expression profiling [12]. We previously described a novel functional enrichment method with high CTC detection sensitivity, which yields sufficient CTCs for downstream applications, such as immunocytochemistry (ICC) profiling [20,21]. In this manuscript, we describe the validation of this technology for use as a BrC detection test. Findings from our case–control and prospective clinical studies show that the test vastly improves CTC detection sensitivity, even in stage 0 BrC (DCIS), and addresses several limitations of prior CTC-based cancer detection efforts.

## 2. Methods

### 2.1. Study Participants and Samples

Samples for method development and validation were obtained from participants in two ongoing observational studies of the sponsor, TRUEBLOOD (http://ctri.nic.in/Clinicaltrials/pmaindet2.php?trialid=31879, accessed on 7 July 2022), and RESOLUTE (http://ctri.nic.in/Clinicaltrials/pmaindet2.php?trialid=30733, accessed on 7 July 2022), the design of which were intended to support the identification and characterization of blood-based malignant-tumor-derived analytes for non-/minimally invasive cancer detection. The TRUEBLOOD study (March 2019—ongoing) enrolls known cases of cancers, as well as individuals with clinical or radiological findings suspected of cancers. The RESOLUTE study (January 2019—ongoing) enrolls asymptomatic adults with no prior diagnosis of cancer, no current symptoms, or findings suspected of cancer and only age associated risk of cancer. Both studies were approved by Datar Cancer Genetics Limited Institutional Ethics Committee (code/registration number—ECR/231/Indt/MH/2015/RR-20), as well as the participating institutes, and were performed in accordance with the Declaration of Helsinki. Fifteen milliliters of peripheral blood were collected from all enrolled study participants in EDTA vacutainers, after obtaining written informed consent. Where possible, tissue samples were also obtained from TRUEBLOOD study participants posted for a biopsy, as per standard of care (SoC) procedures (tissue samples were used for method development). In addition, leftover blood samples from suspected or known (recently diagnosed or pre-treated) cancer patients who availed of the study sponsor’s commercial services for cancer management, as well as healthy (asymptomatic) volunteers at the study sponsor’s organization, were also obtained after due consent. Blood samples (15 mL) from suspected cases of cancers were collected prior to the patients undergoing an invasive biopsy. All biological samples were assigned alphanumeric barcodes, and stored at 2 °C–8 °C during transport to reach the clinical laboratory within 46 h. Sample blinding avoided systematic differences between groups due to (un)known baseline variables that could affect the test findings, and also eliminated potential biases that could have otherwise arisen due to operator’s knowledge of the sample. From the originally collected 15 mL blood samples, a 5 mL aliquot was set aside for processing (CTC enrichment and ICC profiling) as part of clinical studies. The remaining blood samples were used for various method development studies. All samples were processed at the CAP and CLIA-accredited facilities of the study sponsor Datar Cancer Genetics, which also adhere to quality standards ISO 9001:2015, ISO 27001:2013, and ISO 15189:2012. The reporting of observational studies in this manuscript is compliant with STROBE guidelines [22].

### 2.2. Enrichment of Circulating Tumor Cells from Peripheral Blood

Aliquoted blood samples (5 mL) were processed for the enrichment of CTCs from peripheral blood mononuclear cells (PBMC), as described previously [20,21,23]. Comprehensive details are provided in Appendix A, Appendix A, Appendix A.

### 2.3. Immunocytochemistry Profiling of CTCs

The process of ICC profiling of CTCs was as described previously [21]. Comprehensive details are provided in Appendix A. Figure 1 is a schema of the test showing the various steps in CTC enrichment, and identification by ICC profiling for various markers. The decision matrix for assigning samples as positive, equivocal, or negative, based on the findings of ICC profiling, is provided in Figure 2. Numerical thresholds for assigning samples as positive or negative were based on the limit of quantitation (LoQ) studies, as described under analytical validation. A 20% margin was defined to include those samples (assigned as equivocal) where the CTC counts may be lower than this threshold, due to ~20% losses observed during storage and transport (as explained in the section on analyte stability under analytical validation in the Appendix A).

### 2.4. Method Development and Optimization

Comprehensive details of method development and optimization studies are provided in the Appendix A.

### 2.5. Analytical Validation

Analytical validation established the performance characteristics of the test with standard analyte (SKBR3 cells), spiked into healthy donor blood to generate various dilutions (cell densities). These dilutions were processed as per the described procedures (proprietary differentially cytotoxic medium treatment and ICC profiling) to determine the yield of spiked cells. Comprehensive details of analytical validation studies are provided in the Appendix A.

### 2.6. Case–Control Clinical Study

The ability of the test to discern/identify BrC from asymptomatic individuals was initially ascertained and established in a case–control study with 548 females who were recently diagnosed, therapy naïve cases of BrC, and 9632 healthy females with no prior diagnosis of any cancer, no current suspicion of any cancer, and with BIRADS-I on a mammogram, i.e., no evidence of breast cancer (Appendix A). The detailed inclusion and exclusion criteria are provided in Appendix A. Appendix A is a schema of the overall design of clinical studies. Initially, samples in the asymptomatic cohort were randomized into training and test sets in a 70%:30% ratio. The BrC cases were first segregated by stage (0–IV), and the samples per stage were then assigned to training and test sets in a 70%:30% ratio. The training set samples (384 BrC and 6742 cancer-free females) was initially evaluated, with the analysts unblinded to the status of the samples, to determine the concordance between the clinical status and the interpretation of the marker status based on the decision matrix. Then the blinded test set, comprising of 164 BrC and 2890 cancer-free females’ samples, was evaluated to determine the performance characteristics. Subsequently, all training and test samples (BrC and healthy) were shuffled, and a random 30% of samples (with stage-wise for cancer) were selected for analysis as test set iteration 2. This shuffling step was repeated to obtain 20 iterations of the test set. From these iterative 20 sets, median and range of sensitivity, specificity, and accuracy were determined. With about 160 cancer samples (cases) in the test set, and 92% expected sensitivity (better than 85%), the power of the study for determination of sensitivity is expected to be about 0.84. Similarly, with about 2792 asymptomatic samples (controls) in the test set, and an expected specificity of 99.99% (better than 99.8%), the power of the study for determination of specificity is expected to be about 0.90.

### 2.7. Prospective Clinical Study

The performance characteristics of the test were next ascertained and established in a prospective blinded study of 141 individuals with clinical symptoms or radiological findings, who were referred for a biopsy due to suspicion of breast cancer (Appendix A). The detailed inclusion and exclusion criteria are provided in Appendix A. Appendix A is a schema of the overall design of clinical studies. All participants provided blood sample prior to the biopsy. The sponsor was blinded to the diagnosis, i.e., the findings of the histopathological examination (HPE). Samples were prospectively accrued in this study until 24 samples were each obtained for stage 0, I, and II, 20 samples were each accrued for stage III and IV, and 30 samples were accrued for individuals with benign findings. With about 110 cancer cases (across all stages), and an expected sensitivity of 93% (better than 85%), this study design has a power of 0.83. Clinical status of samples (cancer/benign) was revealed to sponsors only after sample analysis was complete and test findings shared with the clinical study investigator. From these samples, performance characteristics, including sensitivity, specificity, and accuracy, were determined, with equivocal findings considered as positive and as negative, respectively.

### 2.8. Molecular Concordance Study

In a combined subset of 61 samples from the case–control and prospective cohorts, where matched tumor tissue and blood samples were available, a molecular concordance study was performed. Tumor tissue DNA (ttDNA) was isolated, and profiled by next-generation sequencing (NGS) using the Ion Proton platform and the Comprehensive Ampliseq Multi (409)-gene Cancer Panel. Simultaneously, PBMCs were isolated from the matched blood samples, and used for CTC enrichment. On the 5th day, genomic DNA (gDNA) isolated from all surviving cells was evaluated by a ddPCR assay specific to the driver mutation on a BioRad QX200 platform. Concordance between tumor tissue and CTCs was determined as the proportion of the latter where the corresponding gene variant was detected by ddPCR.

## 3. Results

### 3.1. Method Development and Optimization

The method development and optimization studies show the viability of multiplexed fluorescence analysis of markers with minimal or no cross-interference of markers, as well as the ability to detect CTCs with much lower marker expression than primary tumor cells or reference cell lines. Additionally the study also shows the capability of the test in detecting CTCs, irrespective of patient age, ethnicity, cancer stage, tumor grade, subtype, or hormone receptor status. The findings of the method development and optimization are provided in the Appendix A.

### 3.2. Analytical Validation

The analytical validation studies establish the analyte stability, and also demonstrate the high sensitivity and specificity of the test, as well as significant linear characteristics in addition to high precision. The sensitivity of the test is not adversely affected by presence of potentially interfering substances, or by controlled variations to operating parameters. The findings of analytical validation that establish these performance characteristics of the test are provided in the Appendix A. The summary of the analytical validation studies is provided in Table 1.

### 3.3. Case–Control Clinical Study

We evaluated the performance characteristics of the test in two clinical studies. In the case–control cross-validation study, the median stage-wise sensitivities are as follows: 70% for stage 0, 89.36% for stage I, 95.74% for stage II, 100% for stage III, 100% for stage IV, and 92.07% overall. In the absence of any positive or equivocal findings in the control (cancer-free and asymptomatic) cohort, the specificity of the test (cancer versus healthy) is 100%. Cancer samples (cases) with equivocal findings are considered as positive for determination of sensitivity and accuracy. Table 2 provides the specificity, as well as median of stage-wise and cumulative sensitivity and accuracy across the 20 iterations. Details of this iteration analysis are provided in Appendix A. Sensitivity and accuracy are also determined with samples with equivocal findings being considered as negative. These findings are presented in Appendix A, which also indicates the stage-wise and cumulative range of sensitivity and accuracy.

Thresholds for sample positivity are determined from the limit of quantitation (LoQ) in the analytical validation study (Appendix A). Lower thresholds are considered sub-optimal and not evaluated. Increasing the thresholds leads to a decrease in the sensitivity of the test for the detection of cancer samples, but with no gain in specificity. Since GATA3+ or GCDFP15+ cells are already undetectable in samples from asymptomatic (healthy) individuals, increasing the thresholds for these markers has no benefit to the specificity (which is already at 100%).

### 3.4. Prospective Clinical Study

The second study was an independently conducted, blinded prospective study. Of the total 141 individuals from whom samples were collected, there are 112 breast cancer cases (stages 0–IV), and 29 cases of various benign breast conditions. There are no samples with equivocal findings in the cancer cohort, hence, the overall sensitivity is 94.6%, with stage-wise sensitivities of 87.5% for stage 0, 95.8% for stage I, 95.8% for stage II, 95.0% for stage III, and 100% for stage IV. Two samples with equivocal findings were diagnosed with benign conditions of the breast. In the absence of follow-up data indicating if these cases were indeed subsequently diagnosed with breast cancer, the samples are considered as false positives (worst-case scenario), based on the specificity of the test (cancer vs. benign), which is determined to be 93.1%. When samples with equivocal findings are considered as negative, the specificity of the test (cancer vs. benign) is 100% (best-case scenario). The sample-wise details of the prospective validation cohort findings are provided in Appendix A. The stage-wise and cumulative sensitivity and accuracy for both these scenarios are provided in Appendix A.

Thresholds for sample positivity in this study are similarly determined from the limit of quantitation (LoQ) in the analytical validation study (Appendix A). Lower thresholds are not evaluated. Among the 29 individuals with benign breast conditions, there are 2 cases with equivocal findings. While higher thresholds may improve the specificity in the benign cohort, they have an adverse effect on the sensitivity for the detection of cancers. In evaluating symptomatic individuals suspected of breast cancer (diagnostic triaging), sensitivity is prioritized to avoid false negatives and improve detection. Hence, greater thresholds to improve specificity (at the cost of sensitivity) are not evaluated.

### 3.5. Molecular Concordance Study

We identified a subset of 61 samples where driver mutations (allele frequency >0.14) are detected by NGS in tumor tissue; for variants detected in 53 samples, a specific TaqMan ddPCR assay is available. A CTC-enriched fraction from these samples is used for gDNA isolation, which, in turn, is evaluated by a ddPCR assay specific to the driver mutation on a BioRad QX200 platform. Variants in ttDNA detected by NGS are also detected by ddPCR in 81.1% of CTCs, indicating significant concordance (Appendix A).

## 4. Discussion

We describe a blood test for BrC detection in asymptomatic women based on multiplexed fluorescence ICC profiling of CTCs in peripheral blood. The test can accurately determine the presence of CTCs in BrC irrespective of stage, grade, subtype, age, ethnicity, or hormone receptor status (Appendix A). Analytical validation establishes high sensitivity, specificity, precision, and robustness, in addition to non-interference from endogenous and exogenous factors (Appendix A). Two separate clinical studies establish 100% specificity (cancer vs. asymptomatic), with 92–94% overall sensitivity and 70–87% stage 0 sensitivity (Table 2). The test can differentiate samples from cancer patients and healthy individuals with high (100%) specificity, and can also identify individuals with benign conditions with ≥93% specificity. Our test has (a) high sensitivity, especially for early stages including DCIS, for more effective detection of cancers at localized stages, which are amenable to curative resection, and (b) high specificity, so that the vast majority of cancer-free individuals do not undergo additional unnecessary procedures. Our test offers compelling advantages over screening mammography and is, hence, a strong candidate for non-invasive BrC screening in asymptomatic women.

Presently, any benefits of standard mammography screening are largely in populations aged 50 years and above who have a higher age-associated cancer risk [24,25,26,27,28,29,30]. Standard 2D digital mammography is reported to have 73–87.3% sensitivity and 86–96% specificity [31,32,33]. The low accuracy of screening mammography is noted in younger women, particularly those aged below 40 years [34]. In addition, challenges associated with screening mammography are the high rates of false positives (7–12% at first mammogram [35] and 50–60% after ten yearly mammograms [36]). Besides, mammography also has a lower sensitivity for invasive cancers (76–85%) than DCIS (83.0–94.3%) [32,33]. Prior studies also suggest a modest association between radiation exposure in mammograms, and elevated risk of cancer in BRCA1 mutation carriers [37].

A longitudinal study of screening mammograms in 69,025 women reports 705 cases of screen-detected BrC (SBC) and 206 cases of interval BrC (IBC, not detected by mammography) [38]; the latter are more likely to be of high-grade, as well as have higher mortality than SBC. Niraula et al. encourage a re-evaluation of the concept of population-based screening mammography, and recommend exploring strategies beyond conventional screening mammography. In support, the 2018 data available at the US National Centre for Health Statistics, Centres for Disease Control and Prevention indicates that approximately 27.1% of women in the age group 50 to 74 years default on SoC mammography [39]. Improved BrC screening and risk-mitigation strategies are, hence, vital to improve compliance and BrC detection.

Recent efforts at developing non-invasive cancer screening technologies focus on a multi- or a pan-cancer approach. Notably, GRAIL’s Galleri introduced the pan-cancer screening test based on methylation profiling in ctDNA [40]. However, the Galleri test has very low sensitivity (<10–16%) for stage I BrC [41,42], with no data on its ability to detect DCIS. Similarly, the CancerSEEK test, based on simultaneous evaluation of serum proteins and gene variants, has ~40% cumulative sensitivity for early stage and overall ~30% sensitivity for BrC [43]. Purposeful screening for early cancer detection necessitates sensitivity for early stages (0–II), which is not demonstrated by these tests. To the best of our knowledge, there are no other platforms with high sensitivity and specificity for early stages (0–II) of BrC.

The present test is based on detection of BrAD-CTCs, which are ubiquitously found in the blood of patients with an underlying breast cancer, and are undetectable in healthy individuals [20,21]. We show that functionally enriched BrAD-CTCs differentiate between breast cancer samples and samples from benign breast conditions, as well as from asymptomatic women with no underlying breast cancer, with high specificity. Obtaining numerically sufficient BrAD-CTCs is akin to a non-invasive biopsy of the breast tumor without stromal or other non-tumor content. Since the present test is based on detection of BrAD-CTCs, which represent the hematogenous phase of carcinomas associated with a higher risk of progression/metastasis, it is likely that the test has a higher sensitivity for detecting those sub-populations of DCIS, where the risk of progression is higher.

Traditionally, epitope capture with Anti-EpCAM is the preferred method for CTC enrichment. However, several studies demonstrate the poor CTC capture/detection rate of this platform [15,16,44]. It would be pertinent to mention that, although technologies like CellSearch are frequently mentioned in research pertaining to cancer detection, these are not approved for the detection of cancers based on CTCs. Hence, the limitations of these technologies for cancer detection must be critically understood and proactively addressed to improve CTC and cancer detection; our test was developed on such a working hypothesis. The label- and size-agnostic functional CTC enrichment technique in our test is immune to the limitations of epitope capture platforms and, hence, may offer a more realistic CTC detection rate. In our test, marker expression is determined by a sensitive high-content-screening (HCS) system, with standardized thresholds to minimize false negatives. The detection thresholds of the test accommodate CTCs with significantly lower marker expression (as compared to tumor cells or reference cell lines), such as those undergoing epithelial to mesenchymal transition [45,46].

The potential benefits of the test include early BrC detection, especially in asymptomatic women who decline guideline-recommended screening mammography, as well as in asymptomatic women for whom the guidelines may not recommend routine screening mammography (e.g., those below 50 years of age). The high (>90%) cumulative sensitivity at stage 0-II indicates a <10% risk of missing these localized cases where the disease has not spread to other organs, and where the 5-year survival rate is ~99%. For the <10% cases that are not detected at local stages, subsequent detection at stage III (regional spread) with >95% reported sensitivity is still associated with ~86% 5-year survival. The test also has a significantly higher sensitivity for invasive carcinomas (all stages) than has been reported for screening mammography [32,33], and can potentially mitigate risks of IBC (this is yet to be prospectively established).

The high specificity of the test translates into a negligible risk of false positives in women without breast cancer. In our case–control study, false positive findings (of BrAD-CTCs) are not observed in blood samples from asymptomatic women with no suspicious findings (BIRADS I) on mammography. The absence of false positive findings in these samples may be attributed to the stringent criteria for (a) BrAD-CTC enrichment, which is based on a hallmark characteristic of cancer, as well as for (b) BrAD-CTC detection, which is based on the positive expression of GATA3 and GCDFP15 in addition to EpCAM and PanCK. There are limited or no risks associated with use of the test, since it is non-invasive and is performed on a venous blood draw of 5 mL of peripheral blood.

The strength of our study stems from the use of an adequately powered sample size and the avoidance of overfitting, since the findings of the iterative validation study agree well with that of the training set.

The test has certain limitations in the context of a universal BrC screening. The sensitivity of the test is lowest for stage 0 disease. However, this does not present any increased risk of false negatives as compared to screening mammography. Since individuals with potentially false negative findings would not be deprived of standard mammography screening, it would not add to the pre-existing risk of the individual. While there is virtually no risk of false positives, the detection of BrAD-CTCs may be construed as false positives in individuals where the malignancy may not be immediately evident on a standard screening mammogram or in a biopsy (as observed in individuals with benign findings in the prospective validation cohort). This risk may be mitigated by use of a more diagnostically relevant imaging modality or follow-up among individuals with positive test findings. Minor non-(Adeno) carcinoma subtypes of breast cancers are not detected by this test. The test has not been evaluated in a prospective large cohort study with the intent to test asymptomatic population. Finally, as inherent to any cancer screening test, our test could result in over-diagnosis and over-treatment.

## 5. Conclusions

We describe a blood-based, non-invasive test that detects breast-AD-associated CTCs with high specificity and sensitivity. The test presents a superior alternative to mammography screening of asymptomatic women for BrC detection. Approximately 38 million mammograms are performed every year in the US [47] of which ~280,000 (~0.75%) of cases are diagnosed with BrC [1,48,49]. Similarly, of the ~16 million mammograms performed annually in Europe [50], ~500,000 [1] (~3.1%) are diagnosed with BrC. Our test has the potential to minimize the need for mammography screening in individuals with positive findings who could be referred for standard assessments, including diagnostic imaging and work up leading to a final confirmed diagnosis. The test may also minimize the need for screening mammography in individuals with negative findings. The test can, thus, improve the accuracy of breast cancer detection.

## Figures and Tables

**Figure 1 cancers-14-03341-f001:**
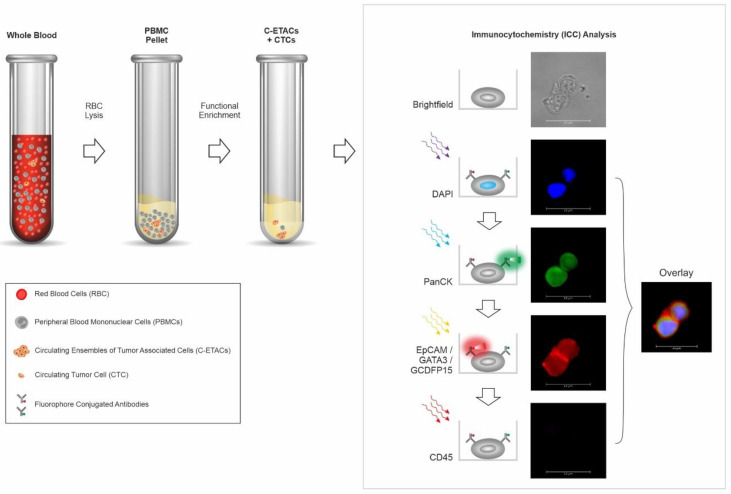
Schema of test. Functional enrichment of circulating tumor cells (CTCs) is achieved using a cell culture medium that is cytotoxic towards all non-malignant cells, and permits survival of tumor-derived malignant cells. Peripheral blood mononuclear cells (PBMC) isolated from whole blood are treated with the medium for 120 h, after which the surviving cells and cell clusters are harvested and evaluated by multiplexed immunocytochemistry (ICC) profiling, to determine presence of breast-adenocarcinoma-associated CTCs (BrAD-CTCs), which are identified as CD45-negative cells that express GATA3, GCDFP15, and EpCAM in combination with PanCK.

**Figure 2 cancers-14-03341-f002:**
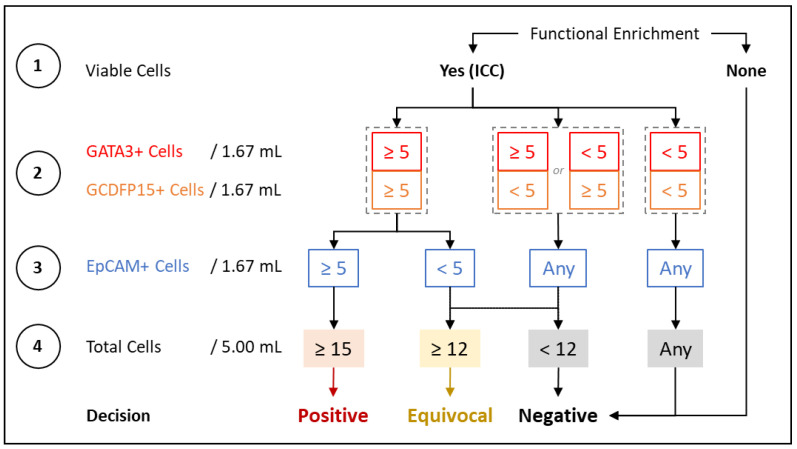
Decision matrix for classifying samples. The detection threshold for breast-adenocarcinoma-associated CTCs (BrAD-CTCs) is ≥ 15 PanCK cells/5 mL, which is constituted by the detection of ≥ 5 GATA3+, PanCK+, and CD45-cells, along with ≥ 5 GCDFP15+, PanCK+, and CD45- cells, as well as ≥ 5 EpCAM+, PanCK+, and CD45-cells in the respective aliquots. Depending on the number of each type of marker positive cells, samples are marked as positive, equivocal or negative. The decision matrix bestows priority to GATA3 and GCDFP15 over EpCAM while classifying samples to ensure specificity for BrAD over other epithelial malignancies where EpCAM+ cells may be detected but breast-specific markers would be absent. Thus, while the test can detect EpCAM+, PanCK+, and CD45-cells, which may be present in various epithelial malignancies, it specifically reports only BrAD-CTCs.

**Table 1 cancers-14-03341-t001:** Summary of analytical validation studies. The summary of findings of the analytical validation studies indicate that the rest provides consistent, accurate, and reproducible results, with little or no interference from routine endogenous or exogenous factors when samples are obtained, stored, and processed under the recommended conditions.

	EpCAM, PanCK, CD45	GATA3, PanCK, CD45	GCDFP15, PanCK, CD45	Overall
**Analyte stability**	48 h			
**Recovery ^1^**	94.6%	86.4%	88.6%	89.9%
**Limit of detection**	1 cell/mL			
**Linear range**	1–64 cells/mL			
**Linearity**	R^2^ ≥ 0.98	R^2^ ≥ 0.98	R^2^ ≥ 0.98	R^2^ ≥ 0.98
**Sensitivity**	96.0% (86.3%–99.5%)	98.0% (89.4%–99.9%)	94.0% (83.5%–98.8%)	94.0% (83.5%–98.8%)
**Specificity**	100.0% (88.4%–100.0%)	100.0% (88.4%–100.0%)	100.0% (88.4%–100.0%)	100.0% (88.4%–100.0%)
**Accuracy**	97.5% (91.3% to 99.7%)	98.8% (93.2% to 99.9%)	96.3% (89.4%–99.2%)	96.3% (89.4%–99.2%)
**Precision**	CV = 4.6%	CV = 3.9%	CV = 3.8%	CV = 4.1%
**Robustness**	CV < 5%			

^1^ Above 10 cells/5 mL as determined from the linearity experiment. Values within parentheses represent 95% CI.

**Table 2 cancers-14-03341-t002:** Summary of clinical validation studies. The table provides the summary of both clinical validation studies. The stringent cross-validation design of the case–control (cancer versu. healthy) study yields a range of sensitivities and accuracies, the median of which are reported along with the 95% confidence interval (CI) for the median. Cancer samples (cases) with equivocal findings are considered as positive for determination of sensitivity and accuracy. The prospective clinical study evaluates the performance of the test among a cohort of symptomatic cases who were eventually diagnosed with breast cancer, or benign conditions of the breast. In this study, benign samples with equivocal findings are considered as false positives for determination of specificity and accuracy. Additional analyses are provided in Appendix A.

	Case–Control Study, Cancer vs. Asymptomatic Specificity: 100.00% (95% CI: 99.87%–100.00%)		Prospective Study, Cancer vs. Benign Specificity: 93.10% (95% CI: 77.23%–99.15%)	
	Sensitivity	Accuracy	Sensitivity	Accuracy
Cumulative	92.07% 95% CI: 91.12%–93.03%	99.57% 95% CI: 99.34%–99.81%	94.64% 95% CI: 88.70%–98.01%	94.33% 95% CI: 89.13%–97.52%
Stage 0	70.00% 95% CI: 34.75%–93.33%	99.90% 95% CI: 99.70%–99.98%	87.50% 95% CI: 67.64%–97.34%	90.57% 95% CI: 79.34%–96.87%
Stage I	89.36% 95% CI: 76.90%–96.45%	99.81% 95% CI: 99.60%–99.94%	95.83% 95% CI: 78.88%–99.89%	94.34% 95% CI: 84.34%–98.82%
Stage II	95.74% 95% CI: 85.46%–99.48%	99.91% 95% CI: 99.75%–99.99%	95.83% 95% CI: 78.88%–99.89%	94.34% 95% CI: 84.34%–98.82%
Stage III	100.0% 95% CI: 88.43%–100.00%	100.0% 95% CI: 99.87%–100.00%	95.00% 95% CI: 75.13%–99.87%	93.88% 95% CI: 83.13%–98.72%
Stage IV	100.0% 95% CI: 88.43%–100.00%	100.0% 95% CI: 99.87%–100.00%	100.00% 95% CI: 83.16%–100.00%	95.92% 95% CI: 86.02%–99.50%

## Data Availability

All relevant data are included in the manuscript and the Appendix A.

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
