# Peer review of "Accurate Screening for Early-Stage Breast Cancer by Detection and Profiling of Circulating Tumor Cells"

_cancers, 2022, doi:10.3390/cancers14143341_

Round 1
Reviewer 1 Report
Crook and Colleagues present the results of the first breast cancer mammographic screening trial implementing a CTC-based test for diagnostic purposes. The Authors applied a method for live CTC and CTC cluster recovery from whole blood samples described in Akolkar et al. Int J Cancer 2020 and showing high sensitivity and specificity on a large population studies including several solid neoplasms. The paper is of great interest to translational and clinical research scientists.
Comments and points to address before paper publication are the following:
Introduction:
Line 13: “Prior studies have also indicated high detection rates of CTCs in blood samples of patients with early‐stage breast cancers.”
The Authors listed some of the works showing data on CTC detection in early breast cancer. Other references should be mentioned and briefly commented:
- First in 2015 Fina et al. reported the frequency of single CTCs (78%) and CTC clusters (79%) detected by morphological analysis upon filtration through a porous membrane of blood samples collected before surgical treatment (https://pubmed.ncbi.nlm.nih.gov/26349664/; images of cytological samples and extended data can be found at https://www.proquest.com/openview/1feca92a387cd94483c11cc97d4c5ad5/1?pq-origsite=gscholar&cbl=51922&diss=y ).
Antigen-dependent capture followed by gene expression analysis also provided encouraging results:
- In 2021 Jakabova et al. obtained a CTC detection of 85% before and 70.5% after the administration of neoadjuvant chemotherapy by assessing the expression of a series of genes coding for CD24, CD44, KRT19, EpCAM, MUC1, MGB, HER2, ESR, PGR, MRP1, MRP2, MRP4, MRP5, MRP7, MDR1, and ERCC1, on the CTC fraction isolated by a filtration-based method (https://pubmed.ncbi.nlm.nih.gov/34345252/).
- In 2022 Fina et al. showed that positivity frequency of 65% are obtained before starting neoadjuvant therapy when assessing the expression of a panel of genes derived from a CTC signature developed from a breast cancer xenograft model in the CTC fraction isolated through immunomagnetic enrichment by EpCAM, HER2 and EGFR (https://pubmed.ncbi.nlm.nih.gov/35216615/ ).
Line 24: “Several prior studies have highlighted the lower performance of epitope capture arising due to its inability to efficiently harvest or detect CTCs with lower expression of EpCAM and PanCK which are the most routinely employed target markers [11–17].”
As shown in the aforementioned papers, epitope-dependent capture is a sensitive method when used in combination with qPCR-based detection with well-designed panels of genes.
Methods:
1. The Authors should declare compliance with STROBE guidelines.
2. The Authors should explain in detail the rationale behind the decisional tree to assign the CTC positivity (*). Decision-making on “Equivocal” cases should be detailed.
Supplementary Methods line 15: antibody clone and brand name/product code should be provided for antobides, e.g. “ (a) Anti-PanCK (1:500), Anti-CD45 (1:500), Anti-EpCAM (1:500), (b) Anti-PanCK (1:500), Anti-CD45 (1:500), Anti-GATA3 (1:4), (c) Anti-PanCK (1:500), Anti-CD45 (1:500), Anti-GCDFP15 (1:2).”
Results:
1. With reference to Comment 2 to the Methods section, the sensitivity and specificity scores obtained when using different cut-off values in the decisional tree should be showed too (*).
2. The median (IQR) number of CTCs obtained per disease stage should be reported in the Results or Supplementary Results section.
3. In Akolkar et al. Int J Cancer 2020 the Authors describe the presence of Circulating ensembles of tumor-associated cells (C-ETACs), including tumor emboli, immune cells and fibroblasts, and probably other cells of non-cancerous origin. They should provide details on the type and frequency of subpopulations of C-ETACs observed in the case series studies by Crook et al.
Discussion:
Implications of using different cut-off values to call as CTC positive a sample and Authors’ comment on the total number of CTCs obtained per mL of blood sample with their technique based on cell selection through a proprietary culture medium as compared to other techniques should be expressed.
Reviewer 2 Report
Article by Timothy Crook et al. dedicated to early-stage breast cancer screening by detection and profiling of circulating tumor cells. In this manuscript, the authors describe a blood test for breast cancer using a test based on the functional enrichment of CTCs using the detection of GCDFP15, GATA3, EpCAM, PanCK and CD45 by the method of multiplex fluorescence immunocytochemistry (ICC). CTCs associated with breast adenocarcinoma (BrAD-CTC) are identified asCD45-negative cells that express GATA3, GCDFP15 and EpCAM in combination with PanCK.
Big questions are raised by the decision Matrix for Classifying Samples proposed by the authors. Figure 2 is not clear. Where did the blood volume of 1.67mL come from? Why exactly this volume? According to the results of the authors, the GATA3 marker has 100% specificity for breast cancer. This marker is also expressed in other types of cancer.
In this paper lacks data clinical material.
The results of the research are not presented convincingly and are not complete.
Round 2
Reviewer 1 Report
Some points were not addressed.
Point 1.
Comment provided in the first round:
Introduction:
Line 13: “Prior studies have also indicated high detection rates of CTCs in blood samples of patients with early‐stage breast cancers.”
The Authors listed some of the works showing data on CTC detection in early breast cancer. Other references should be mentioned and briefly commented:
- First in 2015 Fina et al. reported the frequency of single CTCs (78%) and CTC clusters (79%) detected by morphological analysis upon filtration through a porous membrane of blood samples collected before surgical treatment (https://pubmed.ncbi.nlm.nih.gov/26349664/; images of cytological samples and extended data can be found at https://www.proquest.com/openview/1feca92a387cd94483c11cc97d4c5ad5/1?pq-origsite=gscholar&cbl=51922&diss=y ).
Antigen-dependent capture followed by gene expression analysis also provided encouraging results:
- In 2021 Jakabova et al. obtained a CTC detection of 85% before and 70.5% after the administration of neoadjuvant chemotherapy by assessing the expression of a series of genes coding for CD24, CD44, KRT19, EpCAM, MUC1, MGB, HER2, ESR, PGR, MRP1, MRP2, MRP4, MRP5, MRP7, MDR1, and ERCC1, on the CTC fraction isolated by a filtration-based method (https://pubmed.ncbi.nlm.nih.gov/34345252/).
- In 2022 Fina et al. showed that positivity frequency of 65% are obtained before starting neoadjuvant therapy when assessing the expression of a panel of genes derived from a CTC signature developed from a breast cancer xenograft model in the CTC fraction isolated through immunomagnetic enrichment by EpCAM, HER2 and EGFR (https://pubmed.ncbi.nlm.nih.gov/35216615/ ).
Authors' reply:
The references suggested by the Ld. Reviewer are now cited to provide additional context.
Comment and points to address before publication:
Manuscript was edited by the Authors as follows (line 25):
"Fina et al have reported >78% CTC detection rates in early‐stage breast cancers using a combination of EpCAM, ERBB2, EGFR and MUC2 markers [11]. Using a combination of size‐based separation and gene expression by qPCR, Jakabova et al have shown 85% CTC detection rates in early‐stage breast cancers [12]."
(A) Description of reference 11 is not proper and (B) reference at https://pubmed.ncbi.nlm.nih.gov/35216615/ was not mentioned.
A. Fina et al in 2015 (ref 11) showed that detection rates are 78% when CTCs are isolated using an antigen-independent capture method, i.e. filtration through porous membranes.
B. In 2022 Fina et al obtain 65% positivity through antigen-dependent capture based on the expression of EpCAM, ERBB2 and EGFR (not MUC2) followed by qPCR of a CTC-specific panel of genes. Reference (https://pubmed.ncbi.nlm.nih.gov/35216615/) could be cited following the sentence provided by the Author in line 37: "...with some improvements in sensitivity when epitope capture is used in combination with gene expression profiling [11,12]" - here Fina et al 2022 https://pubmed.ncbi.nlm.nih.gov/35216615/ should be addedd to be thorough.
Point 6.
Comment provided in the first round:
With reference to Comment 2 to the Methods section, the sensitivity and specificity scores obtained when using different cut-off values in the decisional tree should be showed too (*).
Authors' reply
The positivity thresholds (cut-off values) were determined from the Limit of Quantitation (LoQ) of the test as explained in the Manuscript and Supplementary Materials now updated.
Thresholds lower than the LoQ were not considered optimal and hence not evaluated.
Increasing the thresholds leads to a decrease in sensitivity of the test for detection of cancer samples in both cohorts.
However, since GATA3 and GCDFP15 positive cells are undetectable in samples from asymptomatic (healthy) individuals, increasing thresholds does not benefit the specificity (which is already 100%).
In the prospective cohort of 29 individuals with benign breast conditions, there were 2 CTC positive cases. While higher thresholds may have improved specificity, they would have had an adverse effect on the sensitivity.
While evaluating symptomatic individuals suspected of breast cancer, a higher sensitivity is desired so as to avoid false negatives and improve detection.
Comment and points to address before publication:
Although decision made by the Authors is reasonable, data are presented within the scope of a research article, which is still far from application in a diagnostic context. Therefore, explanations and details regarding the Authors' choice of the sensitivity and specificity should be provided to the reader for the sake of clarity and intellectual honesty. A paragraph can be addedd in one of the subsections of the Methods or as a comment in the Discussion.
Reviewer 2 Report
One of the main remarks of this article is the complete absence of results on the number of CTCs in patients with breast cancer. Although the title of the article implies it. There is absolutely no data on the relationship between the number and profile of CTCs with various stages and molecular subtypes of breast cancer.
The practical significance of the work raises big questions. The authors propose an accurate early-stage breast cancer screening using CTC detection based on Epcam, GCDFP15 GATA3, and panCK markers. But, as you know, these markers do not have 100% specificity. They are also expressed in other types of tumors.
In addition, it is known that during the epithelial-mesenchymal transition, CTCs can lose a number of epithelial markers. It has been shown that neoadjuvant therapy in breast cancer leads to an increase in CTC with a sign of stemness without Epcam membrane expression (Heterogeneity of Circulating Tumor Cells in Neoadjuvant Chemotherapy of Breast Cancer Molecules 2018 Mar 22;23(4):727 doi: 10.3390/molecules23040727 PMID: 29565320; PMCID: PMC6017975.).
The complete absence of false positive results is highly questionable. So, in the section Limits of Detection, Quantitation and Blank, the authors write «.. No GATA3+, GCDFP15+ or EpCAM+ cells were detected in the unspiked samples, i.e., no false positives. Thus, the limit of blank (LoB) was determined to be 0 cells / mL. The limit of detection (LoD) was 1 cell / 5 mL».
In the section "Immunocytochemistry Profiling of Circulating Tumor Cells" the authors write «Briefly, CTCs enriched from 5 mL of blood were resuspended in 1500 mL 1x Phosphate Buffered Saline (PBS) and 100 mL aliquots of enriched CTCs seeded into 15 wells. Cells in each well were equivalent to 333 mL blood sample». However, this calculation is not clear. If you do a mathematical calculation, then for each well is equivalent to 333,333 mL of blood
In the section «Detection Thresholds» (SM) the authors write «..Additionally, numerical thresholds were defined as the proportion of cells staining positively for each marker for the acceptance of positive control (PC; >60%) and negative control (NC; <1%).» The threshold for positive control and negative control is not well understood. How was the result evaluated in PC; ≤60% ?
The presented data of the article is still far from being used in a diagnostic context. In order not to mislead readers, I recommend that the authors add to the article explanations and details regarding the choice of markers and thresholds, describing their sensitivity and specificity. Since the decision tree indicates threshold amounts of CTCs with a specific marker profile, the results should reflect the amounts of CTCs in different clinical cases.
Round 3
Reviewer 2 Report
One of the main remarks of this article is the complete absence of results on the number of CTCs in patients with breast cancer. Although the title of the article implies it. There is absolutely no data on the relationship between the number and profile of CTCs with various stages and molecular subtypes of breast cancer.
The authors absolutely did not show the number of CTCs in patients with breast cancer. In doing so, the authors emphasize that the principle of the test for detection of Breast cancer is the co-expression of markers (GATA3+ / GCDFP15+ / EpCAM+ with PanCK+ and CD45-) on cells above a numerical threshold determined by analytical validation (as explained in the supplementary information). It is further clarified that the test does not include enumeration of CTCs beyond positivity threshold as it does not influence the result of the test.
Since the authors indicate the CTC threshold for “Positive”, “Equivocal” and “Negative” in the Decision Matrix for Classifying Samples, I recommend adding the results for phenotypes and the number of CTCs up to the threshold of positivity with a description of the patient's clinic. Of particular interest are the results in the group of patients with benign breast tumors and in the group of patients with stage 0 and I breast cancer. True negative results (with complete absence of any CTC phenotypes) and results with a CTC count <12 should also be distinguished.
